# Optimisation of the Manufacturing Process of Organic-Solvent-Free Omeprazole Enteric Pellets for the Paediatric Population: Full Factorial Design

**DOI:** 10.3390/pharmaceutics15112587

**Published:** 2023-11-04

**Authors:** Khadija Rouaz-El-Hajoui, Encarnación García-Montoya, Andrea López-Urbano, Miquel Romero-Obon, Blanca Chiclana-Rodríguez, Alex Fraschi-Nieto, Anna Nardi-Ricart, Marc Suñé-Pou, Josep María Suñé-Negre, Pilar Pérez-Lozano

**Affiliations:** 1Department of Pharmacy and Pharmaceutical Technology and Physical Chemistry, Faculty of Pharmacy and Food Sciences, University of Barcelona, Av. Joan XXIII, 27-31, 08028 Barcelona, Spain; khadijarouaz@ub.edu (K.R.-E.-H.); andrealopez@ub.edu (A.L.-U.); or chiclana.blanca@ub.edu (B.C.-R.); alexfraschi@ub.edu (A.F.-N.); annanardi@ub.edu (A.N.-R.); marcsune@ub.edu (M.S.-P.); jmsune@ub.edu (J.M.S.-N.); perezlo@ub.edu (P.P.-L.); 2Pharmacotherapy, Pharmacogenetics and Pharmaceutical Technology Research Group, Bellvitge Biomedical Research Institute (IDIBELL), Av. Gran Via de l’Hospitalet, 199-203, 08090 Barcelona, Spain; 3Laboratorios ALMIRALL, Ctra. de Martorell, 41-61, 08740 Sant Andreu de la Barca, Spain; miquel.romero@ub.edu

**Keywords:** omeprazole, pellets, enteric coating, design of experiments and paediatric population

## Abstract

Liquid formulations are mostly used in the paediatric population. However, with certain active pharmaceutical ingredients (APIs), it is very difficult to guarantee quality and stability; this is the case, for example, with omeprazole. Omeprazole is used as a model drug due to the lack of a paediatric formulation meeting gastro-resistance requirements, which remains a challenge today. In this experimental study, the development of enteric polymer-coated pellets is proposed. It is proposed to use aqueous coating dispersions without the use of organic solvents, which are commonly used in fluidised bed coatings. To do this, the design of experiments method is used as a statistical tool for experiment creation and the subsequent analysis of the responses. In particular, this study uses a randomised full factorial design. The mean weight increases of the protective layer and the enteric coating are chosen as factors. Each factor is assigned two levels. Therefore, the design of the used experiments is a 2^2^ + 1 central point. Overall, the obtained pellets can be an alternative to the compounding formulas of omeprazole that are currently used in the paediatric population, which do not meet the gastro-resistance specifications necessary to guarantee the therapeutic efficacy of this active ingredient.

## 1. Introduction

Administering drugs in the paediatric population remains difficult due to the lack of pharmaceutical forms adapted to the needs of paediatric patients. One setting in which this difficulty is evident is hospital pharmacy services, which resort to classical formulation techniques to try to overcome the lack of marketed paediatric medicines. Oral liquid preparations are used as they are the most suitable for such patients; they avoid the need to swallow tablets or capsules and allow simple dosage adjustment according to body weight or body surface area. Generally, these preparations are not subjected to the same quality, safety, and stability tests as marketed medicines and medical devices due to the inability and lack of hospital facility resources to perform these controls. As a result, many medicines are limited with regard to their use in paediatric patients [1,2].

Omeprazole (OME) is a drug with a very evident lack of a paediatric dosage form that meets the stability, safety, and quality requirements demanded by drug regulatory agencies (FDA, EMA, and WHO). OME-compounding formulas used in paediatrics do not conform to their physicochemical, pharmacokinetic, and pharmacodynamic characteristics, and, therefore, the therapeutic effectiveness is directly affected [3,4,5]. OME is a selective and irreversible proton pump inhibitor (PPI) active substance (API). It is widely used in children and adults to treat peptic ulcers, dyspepsia, gastro-oesophageal and laryngopharyngeal reflux, and Zollinger–Ellison syndrome because of its good tolerability and few adverse effects. It is a substituted benzylimidazole derivative in the form of a racemic mixture of two enantiomers. It is a prodrug that is active in acidic media and that reduces acid production in the stomach, relieves symptoms, and promotes gastrointestinal tract healing. Its stability is directly dependent on pH; it remains practically stable in alkaline conditions but degrades rapidly in acidic conditions. It is absorbed in the small intestine; thus, it must be protected from the acidic environment of the stomach. Otherwise, it will not reach its therapeutic target [6,7,8,9,10].

In general, most galenic development is performed non-systematically; to study the effects of a specific factor on a selected response, the levels of each factor are changed separately while keeping all other factors constant. This experimental methodology involves many experiments and is usually influenced by the experience of the researcher and overall is neither an efficient nor cost-effective strategy [11,12]. In this situation, FFD has several advantages over a traditional experimental methodology, such as the provision of the maximum amount of information with the minimum number of experiments, the possibility of increasing the number of factors studied or their levels, and considering possible interactions between factors, simple modelling results, and a thorough search for the optimal response, among others [13,14].

Developing a paediatric dosage form of OME remains a challenge today, as alkaline liquid dispersions are often prepared, which do not ensure the protection of the OME in the gastric environment. Therefore, this experimental study proposes developing very small enteric polymer-coated pellets which can be used to formulate liquid pharmaceutical dosage forms adapted to paediatric patients. OME enteric pellets have been produced using the fluid bed coating technique using three different layers. To determine the optimal conditions of the coating process, the design of experiment (DOE) method was applied, particularly, the full factorial design (FFD) of factors with two levels each and a central point. Our aim was to study the effects of several factors on one or more responses and to find a mathematical model relating the response to the factors [13,15]. As the quality of pharmaceutical products is a prerequisite in any manufacturing process, this study followed the criteria of the quality by design (QbD) method described in the guidelines of the International Conference on Harmonisation (ICH Q8 R2) [16].

## 2. Materials and Methods

### 2.1. Materials

Micronized omeprazole (CAS no. 73590-58-6) was received from Esteve Química, Barcelona, Spain. Lactose monohydrate (CAS no. 10039-26-6), sodium lauryl sulphate (CAS no. 151-21-3), titanium dioxide (CAS no. 13463-67-7), and Talc (CAS no. 14807-96-6) were purchased from Fagron Ibérica SAU, Terrassa, Spain. Vivapur^®^ MCC spheres were purchased from JRS Pharma GmbH & Co. KG, Rosenberg, Germany. Hydroxypropyl methyl cellulose (CAS no. 9004-65-3) and hydroxypropyl cellulose (CAS no. 9004-64-2) were purchased from Shin-Estu Chemical Co., Ltd., Tokyo, Japan. Eudragit^®^ L-30 D-55 was purchased from Evonik Corp., Barcelona, Spain. Triethyl citrate (CAS no. 77-93-0), disodium dihydrogen phosphate (CAS no. 7558-79-4), sodium dihydrogen phosphate (CAS no. 7558-80-7), sodium tetraborate decahydrate (CAS no. 1303-96-4), tribasic sodium phosphate dodecahydrate (CAS no. 10101-89-0), sodium hydroxide (CAS no. 1310-73-2), disodium hydrogen phosphate 12-hydrate (CAS no. 10039-32-4), hydrochloric acid 5 M (CAS no. 7647-01-0), and ethanol 96% (CAS no. 64-17-5) were purchased from PanReac Química S.L.U., Barcelona, Spain.

The water used for the analysis was of MilliQ grade. All used solvents were of analytical grade.

### 2.2. Methods

#### 2.2.1. Full Factorial Design (FFD)

A randomised full factorial design 2^2^ + 1 centre point was used, i.e., 2 factors were studied at 2 levels to optimise the coating process of microcrystalline cellulose inert pellets (MCC). Our aim was to study and identify the relationships between the factors studied and the responses obtained, thus creating a design space that allows the working conditions to be adjusted and optimised. The statistical programme Minitab 21.0 was used for the creation and analysis of the experimental design. The associations between the factors studied, their interactions, and the responses obtained were mathematically described. 

Inert MCC pellets (200 µm in diameter) were coated with three different layers: active ingredient, protective, and enteric. The studied factors were the average weight increase in the pellets after the second (Factor A) and third (Factor B) coatings. For the protective layer, two values of weight increase were set (2% and 6%), as well as for the enteric layer (50% and 100%). A central point was also studied. The scheme of the experiments is shown in Table 1 (which appears in the next paragraph). The protective coating was applied to avoid possible interactions between the OME and the used enteric polymer, Eudragit^®^ L-30 D-55 [17,18]. As these interactions could lead to the degradation of the OME, the weight increase in the protective layer was chosen as a critical factor. Enteric coating, the most critical and relevant for achieving gastro-resistance in the OME, was the second factor in the FFD. The percentages of weight increase were set at the discretion of the researcher.

The evaluation of the omeprazole content, the percentage of gastro-resistance, and the percentage of release were selected as the responses. Assays were performed according to the guidelines of the European Pharmacopoeia (Ph. Eur.) and the United States Pharmacopeia and the National Formulary (USP-NF). 

#### 2.2.2. Preparation of Omeprazole Enteric Pellets

Inert MCC pellets were coated in a fluidised bed (*Glatt AG*) equipped with a bottom spray coating process on a Würster column. They were coated with three successive coating layers: (1) a drug layer, (2) a protective layer, to avoid possible interactions between the first layer and the third layer, and (3) an enteric polymer layer, to protect the omeprazole from the acidic gastric environment. Coating formulations were developed using exclusively aqueous vehicles, thus avoiding the use of organic solvents, which are not recommended in paediatric formulations due to their potential side effects. The non-use of organic solvents and the completion of the coating process in only 3 steps are advantages over other studies on developing enteric formulations of OME for paediatric populations, such as the study by Federica Ronchi et al. [19]. In this study, organic solvents were used to prepare the coating dispersions, and the process was carried out in 5 steps. 

The first coating dispersion was prepared by dissolving disodium phosphate dodecahydrate, lactose monohydrate, and lauryl sulphate in water (in the listed order). Then, omeprazole was dispersed in the above solution and added to a previously prepared aqueous solution of Hypromellose and hydroxypropyl cellulose. The pH was adjusted to 7.5 with a 0.1 N NaOH solution. The second coating solution was prepared by dissolving Hypromellose in water. The third coating dispersion was prepared by dissolving triethyl citrate and a 1 N NaOH solution in Eudragit^®^ L-30 D-55. At the same time, a dispersion of titanium dioxide and talc in water was prepared. This dispersion was added to the solution and kept under constant stirring until complete homogenisation. Coating dispersions 1 and 3 were passed through a 200 µm sieve before coating to avoid possible lumps that could clog the gun. Furthermore, they were kept under continuous and gentle stirring (mechanical stirrer: Heidolph, model Hei-TORQUE CORE) during the whole coating process to avoid the sedimentation of the insoluble components. 

Table 2 specifies the composition of the 3 coating layers and the function of each excipient in the formulation [20]. 

The coating process was carried out in a dark room to avoid the possible degradation of omeprazole by light. The first and second coating layers were successively deposited on the inert MCC pellets to minimise degradation. In the first coating layer, the dispersion was applied until an average pellet weight increase of 25 ± 3% was achieved. Layers 2 and 3 were applied until the average weight increases specified in the FFD were achieved (see Table 3). Before coating with the third layer, the obtained pellets were sieved to avoid possible agglomerates (600 µm sieve). The pellets obtained were sorted by passing them through a 600 µm sieve (agglomerates) and then a 380 µm sieve (fines). Pellets that passed through the 600 µm mesh and were retained in the 380 µm mesh were considered correct. The working conditions for the three coating layers are detailed in the Appendix A.

#### 2.2.3. Characterisation: API and Coated Pellets

##### Determination of Particle Size Distribution (PSD)

The particle size distribution of micronised omeprazole was determined following the general method “2.9.31. Particle size analysis by laser light diffraction” of the Eur. Ph. [21] using a Mastersizer 2000 (Malvern), with the dry basis SCIROCCO 2000 module. The sample was placed in the Scirocco accessory tray, and the method described in Table 4 was followed. 

The PSD of the pellets obtained from the best-performing FFD experiment was also determined. As the omeprazole pellets were larger than 75 µm, we decided to use the sieving method to determine their PSD, following the general method “2.9.38. Particle Size Distribution. Estimation by analytical sieving” of the Eur. Ph. [22]. Theoretically, omeprazole pellets have a PSD of 510 µm, so 4 sieves with different spacings (0.60, 0.5, 0.40, and 0.30) were used. The sieve cascade was placed on a vibrating sieve shaker (CISA). On the top sieve, 10 ± 0.05 (SD) g of omeprazole pellets was placed and kept under vibration at power 10 for 10 min. The sieves and the base were weighed with the fraction of omeprazole pellets retained. The test was carried out in triplicate.

##### Determination of Flow Properties of OME Enteric Pellets

The angle of repose and sliding velocity measurements were conducted to determine the flow properties of omeprazole enteric pellets. Additionally, the Hausner ratio was measured. The tests were performed in accordance with the recommendations outlined in the European Pharmacopoeia monographs “2.9.16. Flowability” [23] and “2.9.36. Powder flow” [24]. 

An ANORSA funnel with reference X5992 and a sheet of millimetre paper were utilised to measure the angle of repose. The funnel was secured in a metal support clamp, with the centre of the millimetre paper positioned just below the lower mouth of the funnel, 7 cm from the paper. The funnel was covered with a piece of paper and filled with the omeprazole enteric pellets. Subsequently, the paper was removed, and the pellets were allowed to fall onto the millimetre paper. If they did not fall out easily, the funnel was gently tapped with a metal spatula until all pellets slid out. The test was conducted in triplicate.

On the other hand, the sliding speed test was carried out using an ANORSA funnel with the reference X7705. For this test, 100 g of omeprazole enteric pellets was weighed, and the mouth of the funnel was covered with paper. The funnel was then filled with the sample, and the paper covering the mouth of the funnel was removed. The time taken for the entire sample to slide down the funnel was recorded. The test was conducted in triplicate.

##### Determination of Coating Uniformity 

A morphological evaluation of the coating uniformity of the enteric layer (outer layer) and protective and API layers (inner layers) was carried out using scanning electron microscopy (SEM). A J-6510 scanning electron microscope was used, with a GATAN ALTO-1000 freezing unit and a backscattered electron detector (EDS). Coated pellets were cut with a scalpel under a magnifying glass and mounted on microscope specimen holders to observe the different coating layers. The samples were coated with a conductive carbon wire and observed after 24 h. They were observed at different magnifications between 120x and 220x. X-ray microanalysis was performed using an EDS detector to determine the elemental composition of each layer. 

##### Determination of API via Infrared Radiation 

A sample of micronised omeprazole was analysed using an IR spectrometer (Thermo Nicolet, Avatar 320 FT-IR, Caldic, Chicago, IL, USA). This determination was used to identify the API. A plot of the results shows the spectrum of the substance, expressing the frequency values in cm^−2^.

##### Differential Scanning Calorimetry (DSC)

The samples of micronised OME and coated pellets were thermally analysed using a differential scanning calorimeter (DSC). The analysed OME enteric pellets were obtained from the best-performing experimental design of the 5 specified in the FFD. Thermograms were obtained using a DSC-822e (Mettler-Toledo, Oakland, CA, USA) under a nitrogen flow rate of 50 mL/min. The samples were crimped in an aluminium sample dish and heated at a rate of 10 °C/min from 30 to 300 °C. Also, the melting point of API was determined. 

##### Determination via X-ray Diffraction (XRD)

XRD analysis was performed using an X’Pert Pro MPD X-ray diffractometer (PANalytical, Malvern, UK). The samples of micronised OME and OME enteric pellets from the best-performing FFD experiment (intact and grounded) were encapsulated between polyester films with thicknesses of 3.6 micrometres. The measurements were carried out from 2 to 60°2θ, with a step size of 0.026°2θ and a measuring time of 300 s per step. 

#### 2.2.4. Evaluation of Omeprazole Content

The technical procedures of European Pharmacopoeia were used as a reference to assess whether the individual omeprazole contents were within the limits set with reference to the average content of coated pellet samples. Ph. Eur. monograph “2.9.6. Uniformity of contents of single-dose preparations” was employed to determine content uniformity [25]. As enteric pellets do not have a specific test, the procedure suitable to tablets was chosen. In this standard, preparation complies with the test if each content is between 85% and 115% of the average content. To assess the omeprazole content of the coated pellets, pellets equivalent to 20 mg OME were weighed and transferred to a 50 mL volumetric flask. Then, 10 mL of ethanol 96° was added, and the flask was sonicated for about 15 min. Next, 20 mL of 0.1 M sodium borate solution was added and sonicated for 15 min. Finally, the solution was tempered and made up to volume with a 0.1 M sodium borate solution. An aliquot was filtered, and the amount of dissolved omeprazole was determined via UV-vis HPLC (Agilent 1100 series, Waldbronn, Germany). The test was performed in triplicate. 

#### 2.2.5. Gastro-Resistance Trial 

The gastro-resistance of OME enteric pellets was determined with a USP apparatus II (Erweka DT 700, Langen, Germany). USP-NF monograph “Omeprazole delayed-release capsules” was used to determine gastro-resistance [26]. For this assay, USP-NF tolerances state that no more than 15% of the amount of omeprazole should be dissolved within 2 h. Each dose, containing coated pellets equivalent to 20 mg of omeprazole, was placed in a vessel containing 0.1 N hydrochloric acid (500 mL) and maintained at 37 ± 0.5 °C with a shaking speed of 100 rpm. Six samples from each FFD experiment were analysed. After 2 h, the medium containing OME enteric pellets was filtered through a sieve with an aperture of NMT 0.2 mm. The samples were collected in a sieve and rinsed with water. With approximately 10 mL of ethanol 96°, OME enteric pellets were carefully transferred to a 50 mL volumetric flask and sonicated for 15 min. After that, 20 mL of 0.1 M sodium borate solution was added and sonicated again for 15 min. Finally, the solution was tempered and made up to volume with 0.1 M sodium borate solution. An aliquot was filtered, and the amount of dissolved omeprazole was determined via UV-vis HPLC (Agilent 1100 series, Waldbronn, Germany). 

#### 2.2.6. Dissolution Trial 

The drug release profiles of OME enteric pellets were determined using a USP apparatus II (Erweka DT 700, Langen, Germany). The dissolution method from the USP-NF monograph for “Omeprazole delayed-release capsules” was used [26]. According to the USP-NF tolerances, no less than 75% of the omeprazole should dissolve within 30 min. Each dose, which contained OME enteric pellets equivalent to 20 mg OME, was placed in an apparatus II vessel containing 0.1 N hydrochloric acid (500 mL) and maintained at 37 ± 0.5 °C with a stirring speed of 100 rpm. After 2 h, 400 mL of 0.235 M dibasic sodium phosphate was added to the 500 mL of 0.1 N hydrochloric acid in the vessel. The pH was adjusted to 6.8 ± 0.5 using 2 N hydrochloric acid or 2 N sodium hydroxide as necessary. The samples were taken at 15, 30, and 45 min and filtered before determining the amount of dissolved omeprazole using UV-vis HPLC (Agilent 1100 Series, Waldbronn, Germany). The dissolution assay was performed in triplicate for each FFD experiment. 

## 3. Results and Discussion

### 3.1. Characterisation: Micronised OME and OME Enteric Pellets 

#### 3.1.1. Particle Size Distribution 

The PSD determination of raw material indicated that 10% of omeprazole particles are smaller than 1.299 µm, 50% are smaller than 4.872 µm, and 90% are smaller than 12.913 µm. It is confirmed that the omeprazole used in this study was micronised (see Appendix A). OME enteric pellets obtained from FFD experiment 4 showed the best gastro-resistance and release results. Therefore, they were chosen for characterisation. The PSD of these pellets indicates that 70% ± 0.68 (SD) have a mean diameter between 0.6 and 0.5 mm. Thus, the theoretical size of the OME enteric pellets is confirmed (see Table 5). The Appendix A show the PSD of micronised omeprazole (see Appendix A).

#### 3.1.2. Determination of Flow Properties of OME Enteric Pellets 

Tests to determine the flow properties of the selected pellets, chosen as the final tests, are outlined in Table 6. An average sliding velocity of 6.05 s ± 0.15 (SD), an average angle of repose of 27.39° ± 0.84 (SD), and a Hausner ratio of 1.086 were obtained. According to the “2.9.36. Powder Flow” [24] monograph of the Ph. Eur., the enteric omeprazole pellets from experiment 4 exhibit excellent flow properties. This is evident as the angle of repose falls within the range of 25–30°, the Hausner ratio falls within the range of 1.00–1.11, and the sliding velocity is notably rapid.

#### 3.1.3. Determination of Coating Uniformity 

Regarding the microscopic observation of OME enteric pellets via SEM, Figure 1 shows the resulting images from secondary electron (SEI) and backscattered electron (BEC) detection. The figure also shows images obtained from the XR microanalysis (EDS) of the elemental composition of pellet coating layers. Thus, the inert core, active layer, and enteric layer are identified. Small imperfections such as roughness, porosity and cracks are identified on the surfaces. These coating imperfections are in line with the results obtained in the gastro-resistance test, in which 95% of the total APIs was recovered and 5% was degraded. The degraded percentage of APIs is due to small parts of the coated pellet surface that were not fully coated by enteric polymer. It is also worth mentioning that these imperfections could be due to the lack of precision when cutting the pellets, which was not easy due to their size. Figure 2 shows the results of EDS mapping, which corroborate the above observations: the inert CCM core is clearly differentiated from the API layer and enteric layer. The protective layer, being so thin and containing only carbon, hydrogen, and oxygen in its elemental composition, could not be differentiated. The obtained images show a remarkable surface homogeneity. Figure 2G shows sulphur, an element that is only part of the chemical structure of omeprazole, homogeneously distributed around the inert core (MCC spheres). This indicates a homogeneous distribution of omeprazole in the obtained coated pellets. Figure 2E,F,H shows magnesium, silicon, and titanium homogeneously distributed in the outermost layer. These elements are part of the chemical structure of excipients in the enteric coating. Figure 2D shows how sodium is distributed in the API and enteric layers, as this element is part of the elemental composition of excipients in both layers (Na_2_PO_4_·12H_2_O in the API layer and NaOH in the enteric layer). 

#### 3.1.4. Infrared Radiation, Differential Scanning Calorimetry, and X-ray Diffraction 

The IR spectrum of micronised omeprazole demonstrates characteristic stretches, which confirms the identity of the API used in the experiments. Appendix A shows the obtained spectrum. In summary, the observed characteristic stretches in the IR spectrum are the following: (I) the absorption band for C=C stretching vibrations of the benzene ring is observed at 3062 cm^−1^; (II) C-H stretching vibrations are observed at 2903.4 cm^−1^; (III) C-N stretching vibrations of the pyridine ring are observed in the range of 1158.54–1310.92 cm^−1^; (IV) N-H bending vibrations of the pyridine ring are observed in the range of 1510.14–1627.12 cm^−1^, and the absorption band for S=O stretching vibrations of the sulfone group is observed in the range of 1012.25–1111.94 cm^−1^.

DSC and X-ray analysis were used to investigate the micronised omeprazole’s physical state. The DSC thermograms revealed that the raw OME material melted at approximately 158.42 °C, a value that aligns with the literature and affirms the crystalline nature of the raw API. In the case of OME enteric pellets, the first phase of melting was attributed to triethyl citrate at 60.08 °C, a value in agreement with the documented melting point of this excipient in the scientific literature [27]. Subsequently, the thermograms exhibited the melting of micronised omeprazole at 143.08 °C, accompanied by two endothermic bands associated with decomposition processes (with peaks at 149.22 °C and 157.95 °C). A third band (with a peak at 206.95 °C) was indicative of the decomposition of Eudragit^®^ L30 D-55 [28,29]. The X-ray diffractogram of micronised omeprazole exhibited its characteristic peaks related to a crystalline structure. The diffractogram of the OME enteric pellets (ground or intact) revealed crystalline excipients, such as titanium dioxide (Antase) and talc. Additionally, amorphous, or partially crystalline, excipients were observed. The peaks of the crystalline phases of omeprazole (using the diffractogram of micronised OME as a reference) were clearly visible. Figure 3 shows the diffractograms and thermograms obtained for both omeprazole and coated pellets. The Appendix A show the individual thermograms and diffractograms for each sample.

### 3.2. Experimental Responses of the FFD 

#### 3.2.1. Evaluation of Omeprazole Content 

The evaluation of the OME content tests was satisfactory: the average API content in OME enteric pellets of the three coating layers of the different experiments was 100% (see Table 7). Therefore, content uniformity complied with the specifications of the Ph. Eur. [25], as the obtained values are within the range of 85–115%. 

#### 3.2.2. Gastro-Resistance Trial 

The degradation of omeprazole in acidic media makes gastro-resistance testing in this medium a prerequisite for demonstrating the stability of the API. 

Table 8 shows the results of gastro-resistance tests performed with the coated pellets obtained from the DoE. The results of experiments 1 and 4 were the most satisfactory, as in both experiments, the USP specification was met, since after the gastro-resistance test, an API percentage higher than 85% is recovered. Experiment 4 is the most optimal, with an average API percentage after the gastro-resistance test of 95%. In experiments 1 and 4, the average weight increase of the enteric coating was 100%, which is the average weight increase necessary to avoid API degradation and comply with the specifications. The reason for the high average weight increase used is that the chosen inert pellets that were used in the experiments are very small (200 µm in diameter). It is necessary to apply a higher amount of the enteric coating to cover the entire surface area of the pellets effectively when dealing with such small particles.

The results obtained in experiments 2, 3, and 5 confirm that with a 50% and 75% increase in the weight of the enteric coating, a level of gastro-resistance exceeding 85% is not achieved. These findings suggest that it is possible that not the entire specific surface area of the pellets is covered with the enteric coating, thus leading to API degradation in an acidic environment. This could explain the behaviours observed in experiments 2, 3, and 5. Additionally, it is important to note that the difficulty in pellet recovery (filtration + transfer to a volumetric flask) after the gastro-resistance test, due to the small size of the pellets, could contribute to the observed losses in the API. Since the results of experiments 1 and 4 confirmed that a 100% increase in the enteric coating weight is required to achieve the necessary gastro-resistance, we concluded that a 100% increase in the weight of the enteric coating is essential in these circumstances to ensure adequate gastro-resistance. 

#### 3.2.3. Dissolution Trial 

Dissolution testing serves as an important tool in the biopharmaceutical characterisation of a product at different dosage stages, from drug development to the quality control and quality assurance of the final product [30]. Therefore, dissolution assays are of great interest in drug development, allowing the simulation of the in vitro behaviour of investigational dosage forms. Dissolution tests were performed on the OME enteric pellets obtained from the DoE. The results of the dissolution profiles are displayed in Figure 4, where experiments 1 and 4 are the fastest-releasing experiments. Both experiments 1 and 4 complied with the USP-NF specifications [26]. After 30 min, they achieved an API percentage of more than 75%, particularly 80.95% ± 2.34 (SD) in experiment 1 and 82.61% ± 1.67 (SD) in experiment 4. On the other hand, experiments 3 and 5 exhibit lower release rates at 67% ± 1.30 (SD) and 65% ± 4.00 (SD), respectively. These two experiments feature a 50% enteric coating. As revealed in the gastro-resistance test (refer to Table 7, Section 3.2.2), this level of enteric coating does not provide complete resistance to the 0.1 N hydrochloric acid medium. Consequently, it can be inferred that the reduced release is attributed to a portion of the API undergoing degradation during the initial phase of the dissolution test. In experiment 2, a slightly higher release rate of 70% ± 1.37 (SD) is achieved.

It is important to emphasise that while a thicker coating typically leads to longer dissolution times, our experimental results reveal a different scenario. Experiments 1 and 4, featuring a 100% weight increase and exhibit a faster dissolution profile compared to experiment 2, with a 75% weight increase, and experiments 3 and 5, with a 50% weight increase. This can be attributed to several factors: Even when formulations are identical, minor variations in coating uniformity and the response to the gastric environment can explain the observed differences in the dissolution rates. Although these differences may be subtle, they hold significant implications for the efficacy of the final product. Furthermore, it is worth noting that the limited degradation of the API during the initial phase of the dissolution test, combined with the physicochemical characteristics of omeprazole, represents a primary factor preventing the 100% release in any experiment.

### 3.3. Statistical Analysis of the Full Factorial Design 

After the regression analysis of the DoE data provided by Minitab, we observed that the evaluation of the omeprazole content depends on both factors, with a *p*-value regression of 0.028 and an R^2^ of 97.3%, and the dissolution depends only on Factor B (average percentage increase of the enteric layer), with a *p*-value regression of 0.066 and an R^2^ of 72%. For gastro-resistance, no model was found that describes its association with the factors studied. The experiment designs did not include replicates, which influenced the statistical analysis results, making them not highly robust. Replicates would provide the possibility to evaluate the existing interactions, and the number of data would allow more precision in determining the model. Therefore, we decided to use statistical analysis in a qualitative way: contour plots to optimise the development process of omeprazole enteric pellets. Figure 5, Figure 6 and Figure 7 show the Pareto and contour plots for each response studied in relation to Factors A and B. In the Pareto diagrams, the factors that exceed the standard line by 85% are the factors that significantly influence each response. In this case, the alpha value used was 0.15, as this avoided discarding data that could have significance in the model studied, allowing for a conservative assessment of the data. The contour plots are also presented, showing the most optimal working areas.

Regarding the evaluation of the omeprazole content response, the optimal working zone corresponds to a percentage increase in the weight of 100% of the enteric layer. However, regarding the protective layer, no differences are observed between using 2% and 6%; all conducted experiments give an omeprazole content higher than 100% (as shown in Figure 5A,B). 

Regarding the gastro-resistance response, according to the Pareto diagram, only Factor B, which represents the enteric coating, is statistically significant for this response (Figure 6A). The contour plot indicates that a combination of 100% of the enteric coating and 2–3% of the protective coating would result in achieving 95% gastro-resistance (Figure 6B). This suggests that the enteric coating has a significant effect on the ability of the pellets to resist degradation in the stomach and ensures that the API is not released prematurely. 

As for the dissolution response, only the protective layer is significant according to the Pareto diagram (Figure 7A). Based on the contour plot, the dissolution response would be compliant with the desired specifications when the enteric coating percentage is higher than 85% and/or when the percentage of the protective layer falls between 2% and 6% (Figure 7B). This combination of enteric layer and protective layer percentages ensures that the drug is released effectively, achieving the desired dissolution profile.

With a 100% average weight increase in the enteric layer and a 2% and/or 6% increase in the protective layer, the best gastro-resistance and dissolution characteristics were achieved. These conditions correspond to experiments 1 and 4 (see Table 3). Although promising results have been obtained in these two experiments, more in-depth research and further experimental studies will be necessary to confirm and verify the findings and explore potential interactions between the factors and studied responses with a more accurate statistical model.

## 4. Conclusions

This study has shown the development of 0.6–0.5 mm diameter omeprazole enteric pellets by applying a full factorial design. The results showed that an optimal coating was achieved using only aqueous coating dispersions, without the use of organic solvents, which has not been published before as far as the authors are aware of. This is of great importance in the paediatric population, as the use of organic solvents in this population is not recommended due to the possible side effects they may cause. We know that due to their morphological characteristics and gastro-resistance properties, OME enteric pellets can be used in pharmaceutical forms for paediatric use as a possible alternative to the compounding formulas of omeprazole currently used in the paediatric population, which must meet the gastro-resistance and quality specifications required to guarantee the therapeutic efficacy of this API. After experimentation, batch 4 is shown to be suitable, which corresponds to the conditions of 2% of the second coating layer and 100% of the third coating layer. The EDS microanalysis of the elemental composition of the inert pellets of experiment 4 of the FFD demonstrated a homogeneity of the coating layers. In the evaluation of the omeprazole content, a percentage of 100% was achieved. In the gastro-resistance test, 95% was not achieved, and in the dissolution test, a release rate of more than 80% was achieved in under 15 min. With these results, Ph. Eur. and USP-NF specifications for omeprazole have been met. Despite the conservative assessment of the statistical analysis results of the FFD, due to its lack of robustness, the proposal of this design is a good strategy to describe the optimal workspace for the two studied factors. Furthermore, the design can also be used to guide further research to optimise the overall coating process.

## Figures and Tables

**Figure 1 pharmaceutics-15-02587-f001:**
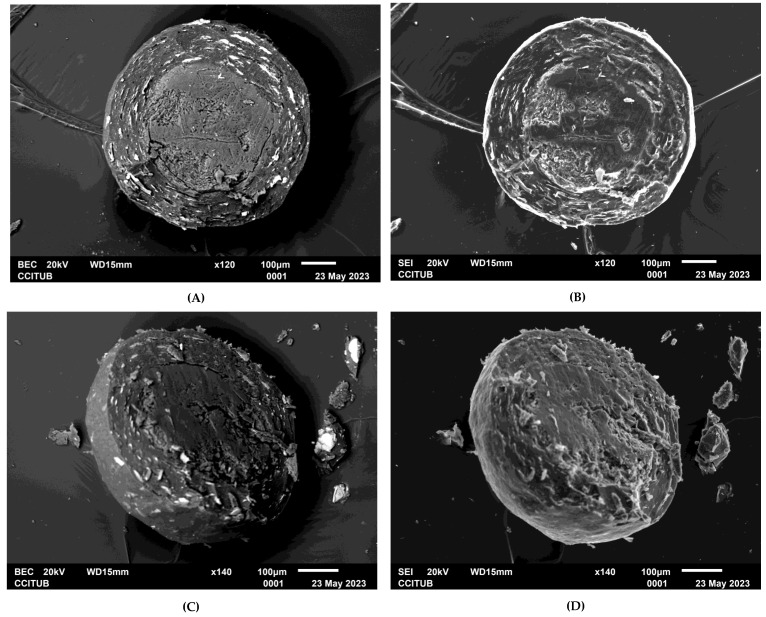
Micrographs of the cross-sectioned enteric pellets of OME obtained via SEM. Images (**A**) and (**B**) were obtained at 120x and (**C**) and (**D**) at 140x.

**Figure 2 pharmaceutics-15-02587-f002:**
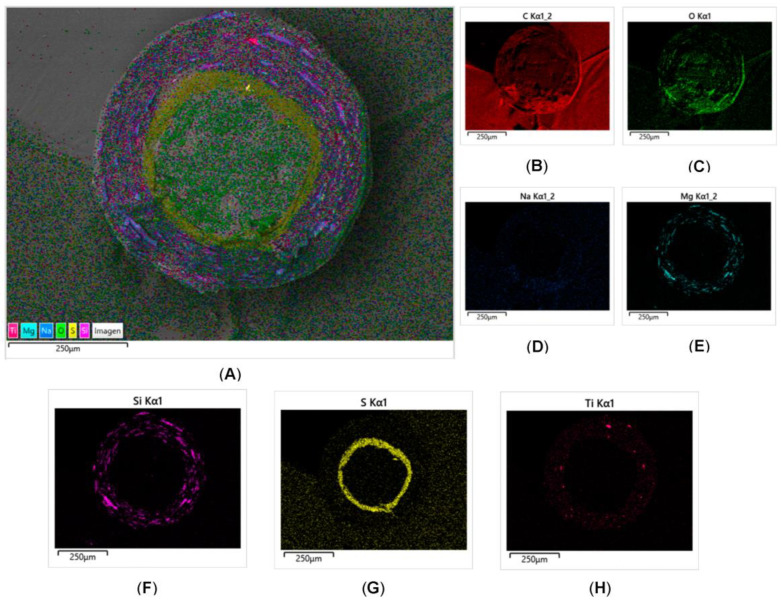
Energy-dispersive X-ray spectroscopy (EDS) micrographs of omeprazole enteric pellets. Subfigure (**A**) presents the results of EDS mapping, illustrating the various components of the coating layers: Ti (pink), Mg (navy blue), sodium (blue), O (green), S (yellow), and Si (purple). Subfigures (**B**–**H**) depict the distribution of each component within the pellets. Notably, (**G**) showcases the homogeneous distribution of sulphur, an element exclusive to the chemical structure of omeprazole, surrounding the inert core.

**Figure 3 pharmaceutics-15-02587-f003:**
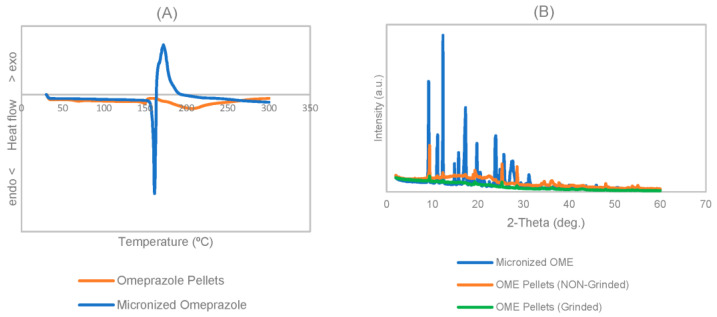
Drug characterisation: (**A**) DSC and (**B**) XRD.

**Figure 4 pharmaceutics-15-02587-f004:**
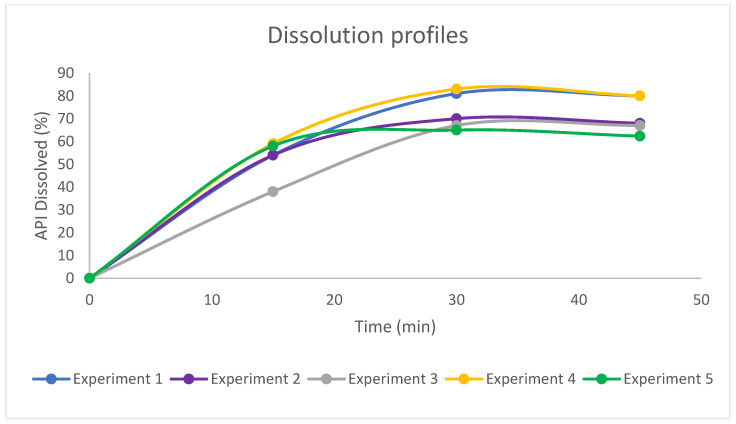
Dissolution profiles of FFD coated pellets.

**Figure 5 pharmaceutics-15-02587-f005:**
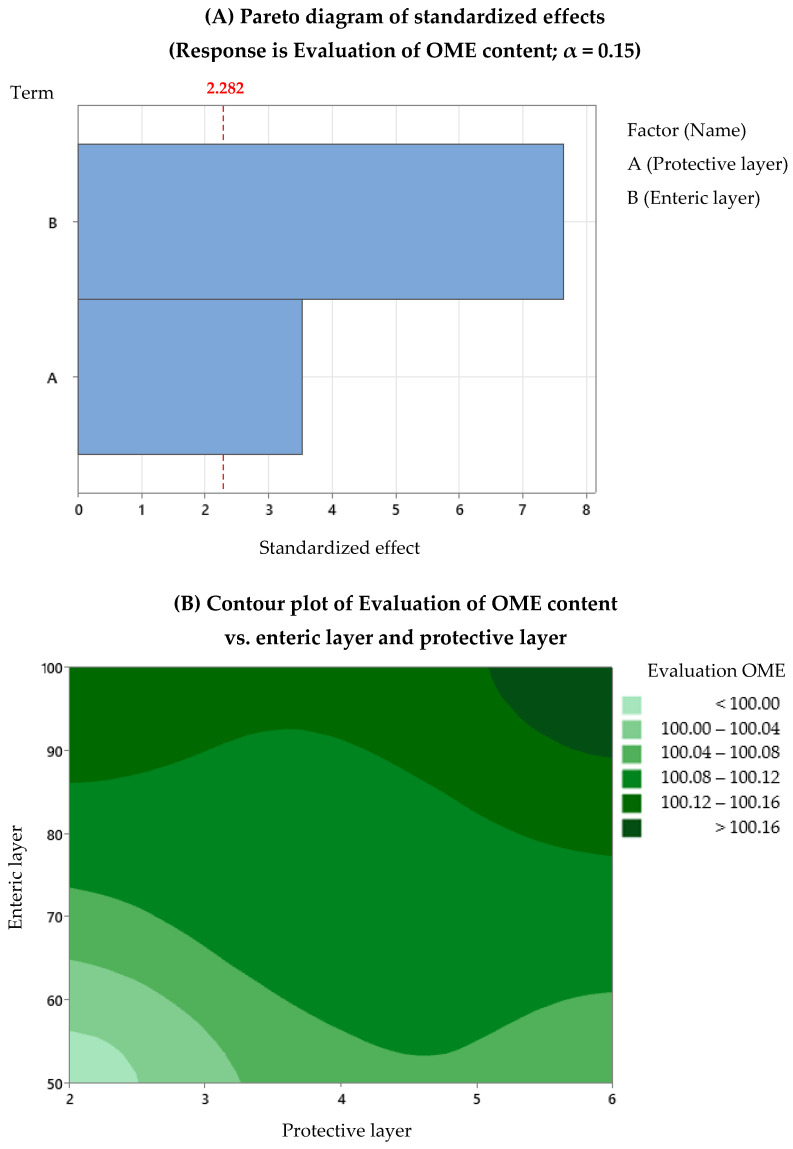
Evaluation of omeprazole content: (**A**) Pareto diagrams and (**B**) contour plots.

**Figure 6 pharmaceutics-15-02587-f006:**
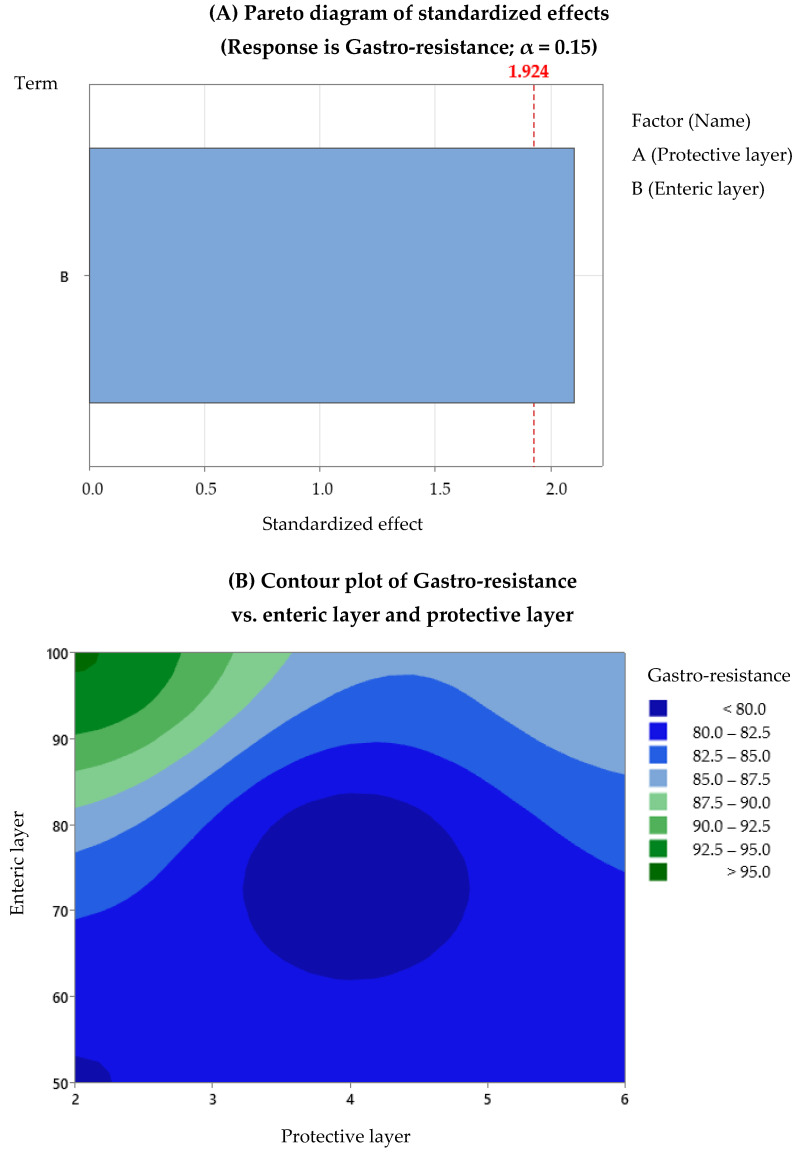
Gastro-resistance: (**A**) Pareto diagrams and (**B**) contour plots.

**Figure 7 pharmaceutics-15-02587-f007:**
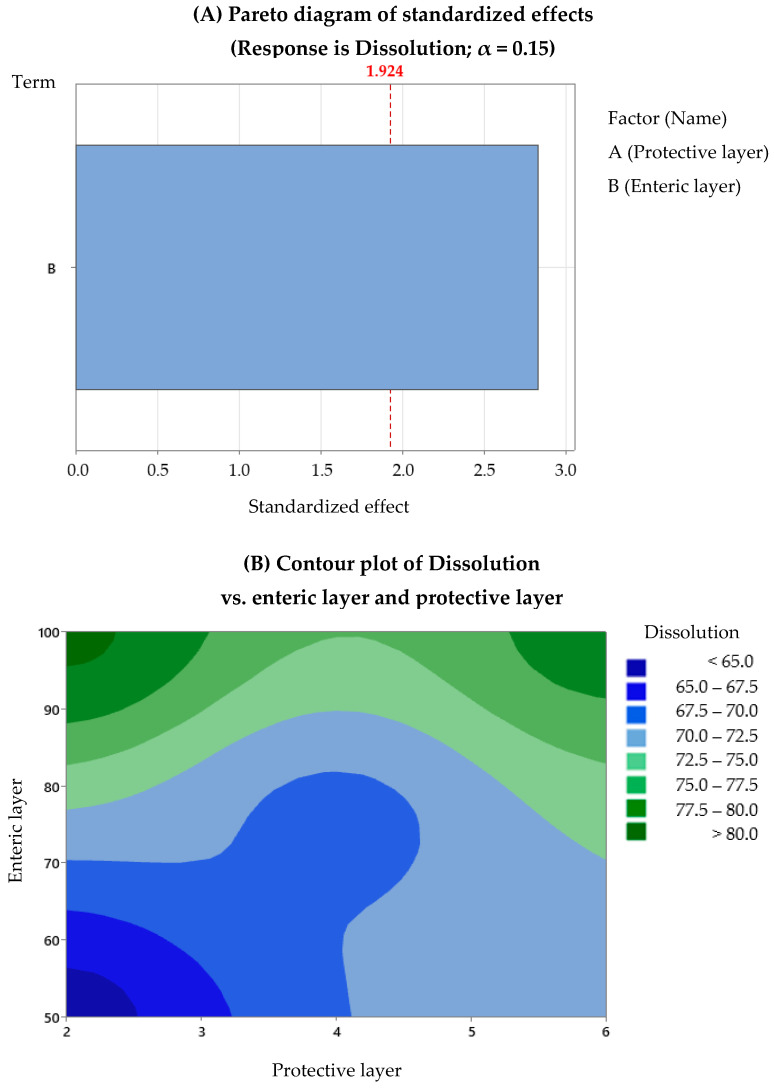
Dissolution: (**A**) Pareto diagrams and (**B**) contour plots. Factor A is not observed in graph (**A**) because the response is solely influenced by Factor B.

**Table 1 pharmaceutics-15-02587-t001:** Design of experiments: randomised full factorial design 2^2^ + 1 central point.

Statistical Order	Running Order	Block	Factor A	Factor B
4	1	1	+	+
5	2	1	0	0
2	3	1	+	-
3	4	1	-	+
1	5	1	-	-

**Table 2 pharmaceutics-15-02587-t002:** Formulations of the coating layers.

Component	Functions	First Coating Layer	Second Coating Layer	Third Coating Layer
Omeprazole, micronised	API	9.50%	---	---
Hypromellose (Grade 606)	Film-forming agent	1.64%	3.40	---
Hydroxypropyl cellulose	Film-forming agent and binder	1.87%	---	---
Disodium phosphate · 12 H_2_O	Buffering agent	0.50%	---	---
Lactose monohydrate	Filler, carrier, and dispersant agent	2.50%	---	---
Sodium lauryl sulphate	Witting and dispersing agent	0.15%	---	---
Eudragit^®^ L-30 D-55	Enteric polymer	---	---	74.56%
Triethyl citrate	Plasticising and film-forming agent	---	---	2.67%
Sodium hydroxide 1 N	pH regulator	---	---	7.23%
Titanium dioxide	Adjuvant of the film-forming agent, opacifier, and pigment blocker	---	---	0.77%
Talc	Opacifying agent	---	---	5.03%

**Table 3 pharmaceutics-15-02587-t003:** Average pellet weight increases according to the FFD.

Experiment	Second Coating Layer: Average Pellet Weight Increase	Third Coating Layer: Average Pellet Weight Increase
1	6%	100%
2	4%	75%
3	6%	50%
4	2%	100%
5	2%	50%

**Table 4 pharmaceutics-15-02587-t004:** Method of determining the PSD of micronised omeprazole.

Material	Polystyrene Latex
Refractive index	1.59
Control of particle distribution	
− Vibration	50%
− Pressure	2 Bar
Measurement cycles	
− Measurement time	6 s
− Measurement snaps	6000
− Background time	6 s
− Aliquot measures	1

**Table 5 pharmaceutics-15-02587-t005:** PSD of OME enteric pellets obtained from FFD experiment 4.

Sieve Light (mm)	Sieve Tare (g) ± SD	Sieve Weight + Retained Sample (g) ± SD	Retained Fraction (g) ± SD	Retained Fraction (%) ± SD
0.60	458.42 ± 0.03	458.42 ± 0.03	0.00	0.00
0.50	433.48 ± 0.05	440.66 ± 0.04	7.18 ± 0.04	70.47 ± 0.68
0.40	381.84 ± 0.05	384.73 ± 0.07	2.89 ± 0.07	28.38 ± 0.59
0.30	407.77 ± 0.11	407.85 ± 0.11	0.08 ± 0.01	0.79 ± 0.09
Base	378.41 ± 0.01	378.41 ± 0.0.1	0.00	0.00

**Table 6 pharmaceutics-15-02587-t006:** Flow properties of OME enteric pellets obtained from FFD experiment 4.

Angle of Repose (°) ± SD	Bulk Density (g/mL) ± SD	Tapped Density (g/mL) ± SD	Sliding Velocity ± SD	Hausner Ratio
27.39 ± 0.84	0.81 ± 0.03	0.87 ± 0.01	6.05 ± 0.15	1.09

**Table 7 pharmaceutics-15-02587-t007:** Evaluation of OME content in coated pellets.

Experiment	Theoretical Dose (mg) ± SD	Actual Dose (mg) ± SD	Dose Accuracy (% (*w*/*w*)) ± SD
Experiment 1	24.67 ± 0.18	24.72 ± 5.49	100.20 ± 1.13
Experiment 2	28.64 ± 3.93	28.67 ± 8.23	100.10 ± 1.41
Experiment 3	27.24 ± 5.32	27.26 ± 7.22	100.07 ± 1.43
Experiment 4	25.19 ± 3.88	25.23 ± 15.40	100.16 ± 2.36
Experiment 5	27.07 ± 10.25	27.06 ± 12.28	99.96 ± 0.40

**Table 8 pharmaceutics-15-02587-t008:** Gastro-resistance of coated pellets.

Experiment	Dose Accuracy after Gastro-Resistance Test (% (*w*/*w*)) ± SD	Amount of API Degraded after Gastro-Resistance Test (%) ± SD
1	87.06 ± 1.06	12.94 ± 1.06
2	78.06 ± 1.88	21.93 ± 2.01
3	80.64 ± 1.34	19.36 ± 1.34
4	95.13 ± 1.29	4.87 ± 1.30
5	79.93 ± 1.59	20.07 ± 1.56

## Data Availability

Data are unavailable due to privacy restrictions.

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
