# Peer review of "Optimisation of the Manufacturing Process of Organic-Solvent-Free Omeprazole Enteric Pellets for the Paediatric Population: Full Factorial Design"

_pharmaceutics, 2023, doi:10.3390/pharmaceutics15112587_

Round 1

Reviewer 1 Report

Comments and Suggestions for Authors

The article describes a successful technological development of omeprazole pellets for use in pediatrics. The article is well written and developed. They use statistical optimization techniques and obtain a technologically adequate formulation.

However, they do not use any type of new technology or material for its development. It is a pharmaceutical development work that lacks novelty. The only difference from other usual developments is that they use smaller pellets to adapt it to children. No in vivo or clinical data are provided. Therefore, even if I consider it a good development work, I do not consider that it has novelty to publish it in a pharmaceutical journal.

Reviewer 2 Report

Comments and Suggestions for Authors

Overall it is a very solid work

I just have one request: Please provide thorough description of the statistical methodology i.e. wat kind of a model was used to obtain the variabls importances - provide its structure and values of coefficients as well.

Reviewer 3 Report

Comments and Suggestions for Authors

The research work by Hajoui et al. presents a novel method for formulating omeprazole pellets for pediatric applications. The use of only aqueous-based coating solutions is a good approach. However, there are a few inconsistencies in the manuscript, which can be categorized as a major revision. The article is suited for publication only after these inconsistencies are addressed.

1.      Section 2.2.1, Line 118: In the text, Table 1 appears in the next paragraph. Edit the sentence suitably.

2.      Section 2.2.1, Line 119: “Protective coating is applied to avoid possible interactions between OME and enteric polymer used.” It is advised to include some references to support the sentence or, if possible, conduct pre-formulation studies to assess the stability of API in the presence of enteric polymer used in the study, and the same can be included in the manuscript.

3.      Particle size analysis report of final granules, if included in the manuscript, could add more weightage to the research.

4.      Figure 3 (B): There seems to be a mismatch in the powder XRD peak position in comparison to the micronized OME from 20° onwards. Powder XRD patterns of excipients are necessary to rule out the possibility of polymorphic phase transitions. If available, the information can be furnished in a supplementary file.

5.      Section 3.1.2. Infrared spectroscopy can only be used only to identify the compounds and not to establish their chemical purity. Line 295 saying “confirms the identity and purity of the API used in the experiments,” could be wrong. Correct the sentence suitably. It is also advised to recheck the manuscript and make necessary corrections to avoid any such claims.

6.      The grades of HPMC used in the study is not mentioned anywhere in the article. Please include the same.

7.      Coated pellets are required to possess adequate flow properties. The authors are required to conduct flowability studies of the prepared pellets and include the results in the manuscript.

8.      Figure 4: There appears to be a typo in the legend of the figure as Experiment 4.1. Correct the same. Also, the units, whether mg or % of API dissolved in ordinate, are not mentioned.

9.      There are some serious concerns with respect to the dissolution study. A few are mentioned below:

a.       From Table 3, it is understood that the layer three thickness is the highest in experiments 1 and 4. It is a common phenomenon that higher coating thickness usually requires longer times to achieve dissolution. However, the results of this study otherwise indicate that the formulation with the highest thickness shows a faster dissolution profile. It is therefore recommended to include a detailed discussion about the dissolution findings and support the findings with suitable references.

b.      The authors are advised to perform the dissolution studies in triplicates to obtain reliable results.

c.       Also explain the reason why the formulations are not able to achieve 100% drug release even after 45 mins of dissolution study.

d.      As per USP monograph for Omeprazole Delayed-Release Capsules, there are two types of tests that can be performed for Omeprazole capsules. Both types involve dissolution testing to be carried out in 2 stages. 1st stage is the acid resistance stage, wherein the amount degraded in 500mL 0.1N HCl (for 2 hours) is quantified, followed by dissolution testing in 900mL of pH 6.8 phosphate buffer. The procedure followed during the dissolution testing in the manuscript is, therefore, wrong and is to be repeated following the correct procedure as mentioned in the USP monograph.

10.  Figure S4 of supplementary file: The DSC Thermograms of micronized API and pellet formulation are not satisfactory. It is therefore advised to provide a detailed discussion of the study. Also, the water loss cannot be confirmed from the DSC study alone. The finding can be supported by a simple water loss study as well.

11.  In Table 5, there appears to be a calculation error in the dose accuracy column. The dose accuracy in the experiment should be ~123%, which is well beyond the spec limit of 115%. A strong justification is required for the discrepancies in the calculation as well as the dose accuracy studies.

12.  Section 3.2.2 Gastro-resistance trial: The findings showing Experiment 1 and 4 with the highest thickness of layer 3 to possess the lowest API loss in 0.1N HCl is acknowledged. However experiments 3 and 5 formulations (with a 50% pellet weight increase) seem to show comparatively less API loss in 0.1N HCl compared to experiment 2 formulation (with a 75% pellet weight increase). Explain the reason for the differences in the discussion section.

13.  The study conducted by the authors seems to be very familiar with the study published by Mummidi et al. in 2018. Hence, the novelty of this research is seriously under question. Link to the published paper: http://dx.doi.org/10.22159/ajpcr.2018.v11i7.25098

14.  Thoroughly recheck the manuscript for any grammatical errors. A few instances have been highlighted:

a.       Line 33, design of experiments instead of design of experiments.

b.      Line 352 and 353 R2 instead of R2.

Comments on the Quality of English Language

NA

Round 2

Reviewer 1 Report

Comments and Suggestions for Authors

I agree with the authors that this is a good job of developing the pellets and an effort to adapt them for administration in children. Although I still consider that it has a reduced novelty, the merit and quality of the paper may be sufficient for its publication.

Reviewer 3 Report

Comments and Suggestions for Authors

The authors have addressed all the raised concerns in the revised manuscript.